# Flavonoids and Terpenoids with PTP-1B Inhibitory Properties from the Infusion of *Salvia amarissima* Ortega

**DOI:** 10.3390/molecules25153530

**Published:** 2020-08-01

**Authors:** Eric Salinas-Arellano, Araceli Pérez-Vásquez, Isabel Rivero-Cruz, Rafael Torres-Colin, Martín González-Andrade, Manuel Rangel-Grimaldo, Rachel Mata

**Affiliations:** 1Facultad de Química, Universidad Nacional Autónoma de México, Mexico City 04510, Mexico; ersalinass@hotmail.com (E.S.-A.); perezva@unam.mx (A.P.-V.); riveroic@unam.mx (I.R.-C.); manuel_erg_p9@hotmail.com (M.R.-G.); 2Instituto de Biología, Universidad Nacional Autónoma de México, Mexico City 04510, Mexico; rafael.torres@ib.unam.mx; 3Facultad de Medicina, Universidad Nacional Autónoma de México, Mexico City 04510, Mexico; martin@bq.unam.mx

**Keywords:** *Salvia amarissima*, PTP-1B activity, amarisolide G, diabetes

## Abstract

An infusion prepared from the aerial parts of *Salvia amarissima* Ortega inhibited the enzyme protein tyrosine phosphatase 1B (PTP-1B) (IC_50_~88 and 33 μg/mL, respectively). Phytochemical analysis of the infusion yielded amarisolide (**1**), 5,6,4′-trihydroxy-7,3′-dimethoxyflavone (**2**), 6-hydroxyluteolin (**3**), rutin (**4**), rosmarinic acid (**5**), isoquercitrin (**6**), pedalitin (**7**) and a new *neo*-clerodane type diterpenoid glucoside, named amarisolide G (**8a**,**b**). Compound **8a**,**b** is a new natural product, and **2**–**6** are reported for the first time for the species. All compounds were tested for their inhibitory activity against PTP-1B; their IC_50_ values ranged from 62.0 to 514.2 μM. The activity was compared to that of ursolic acid (IC_50_ = 29.14 μM). The most active compound was pedalitin (**7**). Docking analysis predicted that compound **7** has higher affinity for the allosteric site of the enzyme. Gas chromatography coupled to mass spectrometry analyses of the essential oils prepared from dried and fresh materials revealed that germacrene D (**15**) and β-selinene (**16**), followed by β-caryophyllene (**13**) and spathulenol (**17**) were their major components. An ultra-high performance liquid chromatography coupled to mass spectrometry method was developed and validated to quantify amarisolide (**1**) in the ethyl acetate soluble fraction of the infusion of *S. amarissima*.

## 1. Introduction

Type 2 diabetes mellitus is a metabolic disease characterized by chronic hyperglycemia due to insulin resistance, or the relative absence of the hormone. The prevalence of the disease is continuously increasing, with approximately 463 million people living with diabetes nowadays. Mexico is one of the countries more affected by type 2 diabetes mellitus, with more than 12 million cases. The population sector more affected is the indigenous people owing to variations in its traditional way of life and the effects of industrial developments [1]. Mexican population employs more than 300 plant species to treat the symptoms of diabetes; in some cases, the patients combine allopathic therapies with the botanical remedies [2]. These plants are an essential part of the country’s alternative medical care, and the best testimony of their efficacy is their persistence in Mexican markets and other places for crude or fresh drug selling. Therefore, it is crucial to analyze these plants to establish their composition, security, efficacy, and to develop a suitable methodology for quality control of the crude drug following good practice guidelines.

Quality control of herbal drugs is the base for their efficacy and safety. Quality control of herbal drugs aims to define their identity, purity and content of active principles or marker compounds. The chemical composition of plants, and hence of their therapeutic preparations, is variable, so standardization is necessary to guarantee comparable therapeutic effects. To prove the constant composition of herbal preparations, there are appropriate standard analytical methods to establish relevant criteria for uniformity. Standard analytical techniques include, among others, high-performance liquid chromatography. For many years, the World Health Organization (WHO) has encouraged all its country members to elaborate pharmacopeic monographs providing comprehensive scientific information on the quality of their most commonly used medicinal plants. Following WHO guidelines, Mexico has developed the Mexican Herbal Pharmacopeia, which contained monographs with definitions, analytical techniques for identity and composition, as well as storage regulations of the most widely used Mexican Herbal drugs [2,3].

Like other pharmaceutical products, herbal drugs should fulfill the basic requirements of being efficacious and safe. To establish herbal drugs’ efficacy and safety is necessary to perform preclinical and clinical assays, including those of the healer or medical doctor in rural communities who apply locally produced herbal preparations. When assessing the efficacy of the plants, it is essential to study both the traditional preparations and their components; this is because sometimes the efficacy is attained with the combinations of compounds in the preparations, which may be acting through synergy, network pharmacology or by targeting several nonrelated proteins involved in the pathology of a disease. Altogether, these studies can lead to the development of standardized phytomedicines of good quality and discover good drug candidates or molecules useful for lead optimization or even fragment-based drug discovery [3]. 

Among the species highly valued in Mexico for treating diabetes is *Salvia amarissima* Ortega (syn. *Salvia circinata* Cav.) belonging to the mint family. It is a perennial aromatic shrub native to Mexico, listed as medicinal in the catalog of plants from the Royal Botanical Expeditions to New Spain. Like many other New World *Salvia* species, *S. amarissima* is melittophilous (bee-pollinated). A tea brewed from dried aerial parts of the plant is useful in Mexican folk medicine for treating diabetes, ulcers and helminthiases [4,5]. The species is commonly regarded as “insulina” (insulin), referring to its efficacy to improve the diabetic condition [5]. Previous phytochemical studies allowed the isolation of some *neo*-clerodane diterpenoids, including amarisolide (1) [6], a few *seco*-clerodane diterpenoids [6,7,8,9,10] and some flavonoids [6,8]. The traditional preparation of the plant collected in Puebla, Mexico, as well as amarisolide (1), and some of the flavonoids showed inhibitory activity against mammal α-glucosidases in vitro and in vivo. The preparation and compounds were hypoglycemic and reduced the postprandial peak significantly during an oral sucrose tolerance test in healthy mice [8]. Some of the *seco*-clerodane diterpenoids were cytotoxic against a few human cancer cell lines, had modulatory activity in a breast cancer cell line resistant to vinblastine and exhibited antiprotozoal action [7,9,10,11]. Furthermore, the traditional preparation lack toxicity when tested according to the Lorke criteria. [8]. More recently, the antinociceptive properties of an aqueous extract of the plant, **1** and **7** were demonstrated [12]. 

Based on the above considerations, this investigation aimed: (i) to determine the effect of the traditional preparation (infusion) of the plant collected in Oaxaca and their components on the activity of the protein tyrosine phosphatase (PTP-1B) in order to assess a new molecular target, and get a better insight in the in vivo hypoglycemic effect previously demonstrated [8]. This target was chosen, considering that this enzyme acts as a negative regulator of insulin and leptin dependent signal cascades holding therapeutic utility in type 2 diabetes mellitus and obesity [13]. (ii) To analyze the chemical composition of the essential oil. (iii) To set up an appropriate procedure using Ultra-High-Performance Liquid Chromatography (UHPLC) to quantify one active or marker compound of the plant’s infusion. The chemical composition of the plant’s essential oil and the UHPLC procedure will allow developing a pharmacopeic monograph of *S. amarissima*, as they represent valuable identity and composition tests, respectively.

## 2. Results and Discussion

Scheme 1 summarizes the whole work.

### 2.1. Chemical Constituents of the Aqueous Extract

An aqueous extract (AE) from *S. amarissima* inhibited the PTP-1B activity significantly, with an IC_50_ value of 88.6 ± 5.4 μg/mL. Workup of AE by solvent partitioning and repeated chromatography afforded eight compounds (Figure 1), namely amarisolide (**1**), 5,6,4′-trihydroxy-7,3′-dimethoxyflavone (**2**), 6-hydroxyluteolin (**3**), rutin (**4**), rosmarinic acid (**5**), isoquercitrin (**6**), pedalitin (**7**) and a *neo*-clerodane type diterpenoid glycoside, named amarisolide G (**8a**,**b**). Compound **8a**,**b**, is a new natural product characterized by conventional spectroscopic and spectrometric techniques. The known compounds were identified by comparing their spectroscopic data with those previously described (Appendix A) [8,14,15]. Compounds **2**–**6** are reported for the first time for this species. In addition, thin layer chromatographic (TLC) analysis revealed that these compounds were present in the AE prepared from the fresh material.

Product **8a**,**b** was obtained as a white solid optically active. DART-HRMS (Direct Analysis In Real Time-High Resolution Mass Spectrometry) established its molecular formula as C_26_H_36_O_11_. The IR spectrum included bands for hydroxyl (3365 cm^−1^) and α,β-unsaturated-γ-lactone (1749 cm^−1^) functionalities (Appendix A) [8]. The NMR spectra of **8a**,**b** (Table 1; Appendix A) were closely similar to those of amarisolide D (**8c**), a *neo*-clerodane type of diterpenoid with an α,β-unsaturated-γ-lactone at C-4/C-5, a five-membered keto-γ-lactol methyl ether and a β-D-glucopyranosyloxy moiety at C-2 [8]. Thus, in compound **8a**,**b** the signals for the five-membered keto-γ-lactol methyl ether were replaced by those of a keto-γ-lactol moiety. Accordingly, the NMR spectra showed signals at δ_H_ 5.90, 6.05 (brs, H-14)/δ_C_ 99.4, 99.9 (C-14), δ_H_ 5.96 (brs, H-16)/δ_C_ 116.1 (C-16), δ_C_ 171.4 (C-13) and δ_C_ 171.9 (C-15) (Table 1). Since the resonances for H-14 appeared as two separate signals, the intensity of each corresponding to one-half proton, compound **8a**,**b** is a 1:1 mixture of C-14 epimers. The doubling of C-14 signal in the ^13^C-NMR spectra provided further evidenced (Table 1). The ^13^C-NMR chemical shifts (Table 1) of C-11-C-16 and the key HMBC (Heteronuclear Multiple Bond Correlation) correlations of H-12 and H-14 established the point of attachment of the ethyl fragment (C-11-C-12) to the keto−γ-lactol ring. The most relevant HMBC correlations were H-12a (δ_H_ 1.72) with C-14 (δ_C_ 99.4 and 99.9); H-12b (δ_H_ 1.56) with C-14 (δ_C_ 99.4 and 99.9); and H-14 (δ_H_ 5.90 and 6.05) with C-12 (δ_C_ 33.8). The NOESY (Nuclear Overhauser Effect Spectroscopy) interactions revealed that the relative configuration at the stereogenic centers of **8a**,**b** was identical to that of amarisolides A–D [8]. The electronic circular dichroism spectrum of **8a**,**b** showed negative Cotton effects at ~212 and ~250 nm due to the electronic transitions π→π* and *n*→π*, respectively, of the α,β-unsaturated-γ-lactone. The latter data indicated that the absolute configuration at the stereogenic centers C-2, C-5, C-8, C-9 and C-10 of compound **8a**,**b** was *S*, *S*, *R*, *R* and *R*, respectively. The D configuration of the β-glucopyranosyloxy moiety was established as previously described [8]. On the basis of these evidences, compound **8a**,**b** was characterized as (2*S*,5*S,*8*R*,9*R*,10*R*,14*R,S*)-2-(*O*-β-d-glucopyranosyl)-*neo-*clerodan-14-hydroxy-3, 13-diene-14,15;18,19-diolide (**8a**,**b**) and was designated with the trivial name of amarisolide G.

### 2.2. Chemical Constituents of the Essential Oil 

Dried and fresh plant materials were analyzed to assess any change during the drying process. Since it is an aromatic plant, the chemical profile of the essential oil is valuable as an identity test. The essential oil content of distilled aerial parts was 0.02% and 0.03% for fresh and dried material, respectively. In each case, eight major compounds were identified, representing 99.96 and 99.97% of the composition of the essential oil, respectively. As observed in Table 2 and Appendix A, the major components in both samples were germacrene D (**15**) and β-selinene (**16**), followed by β-caryophyllene (**13**) and spathulenol (**17**). The only mutually exclusive constituents were 3-methoxy-*p*-cymene (**9**) present in the dried material, and δ-elemene (**11**) found only in the fresh plant. These differences are not due to seasonal changes because the plant material was from the same batch. 

It is worth mentioning that the oils from other *Salvia* species analyzed also contains sesquiterpenes. In this context, *Salvia ceratophylla, S. aethiopis* L., *S. palaestina* Bentham and *S. xanthocheila* Boiss. ex Benth., are rich sources of germacrene D (**15**) [16]; β-caryophyllene (**13**) is the major component of *S. nemorosa* L., *S. verticillata* L.*, S. virgata* Ortega and *S. hydrangea* DC. ex Benth. Finally, germacrene B was the major compound of *S. syriaca* L. [17,18].

### 2.3. Evaluation of Compounds ***1**–**8a,b*** on the PTP-1B Inhibitory Activity

One of the major causes of type 2 diabetes mellitus is insulin resistance, which occurs when the hormone cannot activate signaling pathways in major metabolic tissues (muscles, fat and liver). Insulin resistance involves several inhibitory molecules that interfere with tyrosine phosphorylation of the insulin receptor. Among those, PTP-1B is a promising therapeutic target since it inactivates crucial signaling effectors in the insulin- and leptin-signaling cascades by dephosphorylating their tyrosine residues [13]. Therefore, natural products **1**–**8a,b** were tested against PTP-1B (Table 3). Among the flavonoids tested, the most active component was pedalitin (**7**) with an IC_50_ of 62.0 ± 4.1 μM (19.0 μg/mL), whereas of the diterpenoids was amarisolide (**1**) (279.9 ± 26.0 μM; 137.9 μg/mL). In both cases, the inhibitory effect was concentration-dependent. The inhibitory activity against PTP-1B reported for compounds **4** and **6** correlated well with that found in this work [19,20]. On the other hand, rosmarinic acid (**5**) is an ingredient of an active extract of *Rosmarinus officinalis* against PTP-1B [21]. However, in this study we report for the first time the effect of the pure **5**. The activity of the isolates **1**–**8a,b** was compared to that of ursolic acid (**UA**, IC_50_ = 28.1 ± 1.2 μM), which in other studies displayed lower IC_50_ values (~3.1 μM) [22]. 

The results of the PTP-1B are significant. They suggest that the traditional preparation of *S. amarissima,* with hypoglycemic and antihyperglycemic properties demonstrated in vivo [8] (i.e., the overall action), contains compounds such as **1**–**8a**,**b** that might weakly target different proteins (i.e., PTP-1B and others) within the same signaling network thus shutting insulin signaling cascade process by network pharmacology. It is also possible that compounds **1**–**6** and **8a**,**b**, with weaker activity than compound **7**, altogether put forth a biochemical effect by synergism (i.e., a synergy between weakly active compounds against PTP-1B). Finally, molecules like compounds **1**, **4**, **5** and **7** can exert their action binding different targets such as PTP-1B and α-glycosidases, among others (polypharmacology). The fact that rutin (**4**) [23] and rosmarinic acid (**5**) [21] are multitarget antidiabetic compounds, and compounds **1** and **7** inhibited α-glycosidases in vivo [8] strengthen any of these possibilities.

### 2.4. Docking Study

To predict the preferred binding orientation of compounds **1, UA** and **7** into PTP-1B**,** we performed a docking analysis. Compound **1** was not among the most active compounds but included for comparative purposes. These substances **1**, **7** and **UA** were docked with the co-crystallized structure of PTP-1B at the catalytic (PDB ID: 1G7F) and allosteric (PDB ID: 1T49) sites. The results in Figure 2 and Table 4 indicate that all ligands bind at the same site as the co-crystallized ligands 892 (3-(3,5-dibromo-4-hydroxy-benzoyl)-2-ethyl-benzofuran-6-sulfonic acid (4-sulfamoyl-phenyl)-amide) and INZ (2-{4-[(2s)-2-[({[(1s)-1-carboxy-2-phenylethyl]amino}carbonyl)amino]-3-oxo-3-(pentylamino)propyl]phenoxy}malonic acid) [24,25]. The estimated energy binding is different for each compound. Compound **1** has a higher affinity for the catalytic site, which is more and conserved site, while compound **7** and **UA** targeted its more hydrophobic and less conserved allosteric site. The RMSD values for ligands **1** and **7** are in the range of 2 to 3 Å, which indicates that the ligands do not precisely overlap but maintain the correct orientation [26,27]. The amino acids interacting with compounds **1** and **7**, as well as **AU** are similar to those previously reported for other inhibitors (Table 4; Appendix A) [24,25]. At the allosteric site, compound **7** has hydrophobic interactions with Ala189, Leu192, Phe280 and Phe196 while **AU** with Leu192, Phe280 and Phe196. Therefore, pedalitin (**7**) behaves as **UA** regarding its higher affinity for the allosteric site. For **UA**, the key structural feature is the pentacyclic core displaying a nonpolar characteristic, which interacts with nonpolar residues in the allosteric site [28]. For compound **7**, apparently beside the tricyclic structure, the lack of oxygen at C-3 of the flavonoid core seems to be essential. It will be necessary to pursue a kinetic analysis to determine if experimentally compound **7** is an allosteric inhibitor of PTP-1B. It is essential to mention, that the less-conserved PTP-1B allosteric site is an ideal target for a compound to inhibit PTP-1B activity because the problems associated with inhibition at catalytic site will disappear. Thus, this study may provide an important lead among flavones for the development of novel PTP-1B allosteric inhibitors.

### 2.5. Molecular Dynamics of PTP-1B-Compounds ***1, 7*** and ***UA*** Complexes

Molecular dynamics (MD) studies of the complexes PTP-1B-compounds **1**, **7** and **UA** were carried out to evaluate the stability of the docked complexes illustrate in Figure 3. Table 5 shows the theoretical parameters calculated from 100 ns of MD. All PTP-1B-compound complexes show negative ∆G (affinity parameter) consistent with their stability. Compound **1** has a ∆G similar to **UA** at the catalytic site, while compound **7** has a lower ∆G than **AU** at the same site. Figure 3 and Figure 4 show the structural models, RMSD and Root Mean Square Fluctuations (RMSF) of the molecular trajectories for compounds **1** and **7**, respectively. The RMSD of the complexes is lower with respect to PTP-1B in both the catalytic and allosteric MDs, which indicates the conformational stability of the complexes. In the RMSF analysis, it is observed an area between residues 27–50 (corresponding to a loop), which is stabilized with the ligands. The data obtained with the MDs are complementary and conscious with the docking data and experimental results.

### 2.6. Drug Likeness for Compounds ***1**, **7*** and ***UA***

According to the SwissTargetPrediction (http://www.swisstargetprediction.ch/index.php), and Molinspiration (http://www.molinspiration.com/cgi-bin/properties) databases which predict the most probable targets of small bioactive molecules, compounds **1**, **7** and **UA 1**, **7** could target any protein with percentages of 26.7, 40 and 40 %, respectively. SwissTargetPrediction predicted PTP-1B inhibition for these compounds with probabilities of 0, 0.1266 and 0.95, respectively. According to these predictions, **UA** should have been 7.5 times more active than compound **7**; however, experimentally, **UA** was only 2.2 times more than compound **7** (Appendix A).

Next, using the Osiris Property Explorer server (http://cheminformatics.ch/propertyExplorer), relevant properties for compounds **1**, **7** and **UA** were calculated and summarized in Table 6. These properties indicate whether a molecule is a potential drug. The logP value is a measure of a compound’s hydrophilicity. Low hydrophilicity and, therefore, high logP values cause poor absorption. For a compound being well absorbed, its logP value must not be greater than 5.0; logP values between 1.35 and 1.8 indicate perfect oral and intestinal absorption. Thus, compounds **1** and **7** could have proper absorption, but not **UA**.

The aqueous solubility of a compound influences its absorption and distribution characteristics; a low solubility goes along with inadequate absorption. More than 80% of the drugs on the market have a logS value higher than −4. Compounds **1** and **UA** present −3.48 and −6.11, respectively. The drug-likeness parameter is a complex balance of various molecular properties and structural features that determine whether a molecule is similar to the known drugs. These properties, mainly hydrophobicity, electronic distribution, hydrogen bonding characteristics, molecule size and flexibility, and of course presence of various pharmacophoric features influence the behavior of molecule in a living organism, including bioavailability, transport properties, affinity to proteins, reactivity, toxicity, metabolic stability and many others. A positive value indicates that a molecule contains predominantly fragments, which are frequently present in commercial drugs [29]; compound **7** has a drug-likeness value of 1.8. The H bond acceptor and H bond donor’s parameters indicate a molecule’s ability to interact to a greater or lesser degree with a protein; compounds **1** and **7** present a higher number of possible hydrogen bridges than **UA**. Finally, the drug-score is an indicator that qualifies the potential of a compound for being a drug based on all the calculated parameters; compound **7** has the best drug-score (0.52), which is in harmony with the experimental and theoretical data. 

### 2.7. Development and Validation of an UHPLC-MS Method for Quantifying ***1***

Initial assessments about the complexity of samples of AE were based on visual comparisons of their chromatographic profiles. The ethyl acetate soluble fraction of AE yielded the best profile. Chromatographic separation was performed on Acquity UHPLC^®^ BEH Shield C_18_ column (2.1 × 100 mm, 1.7 μm) applying a binary gradient elution of water (0.1% formic acid) and MeCN. The total run time was 10 min. As illustrated in Figure 5 compounds **1**, **2** and **4**–**7** are present in the chromatogram. All compounds were identified by their retention times and *m/z* values corresponding to [M − H]^−^ ions. Compounds **1**, **2** and **4**–**7** showed an effective baseline resolution. The pseudo molecular ions of these compounds appeared at *m/z* 609.54 [M − H]^−^ (**4**; R_T_ 1.36 min), 463.46 [M − H]^−^ (**6**; R_T_ 1.60 min), 359.23 [M − H]^−^
**(5**; R_T_ 2.46 min), 315.48 [M − H]^−^ (**7**; R_T_ 2.82 min), 329.70 [M − H]^−^ (**2**; R_T_ 3.65 min) and 491.23 [M − H]^−^ (**1**; R_T_ 4.89 min). The main component of the ethyl acetate fraction was amarisolide (**1**), then selected as a marker for validation. So far amarisolide (**1**) has been only isolated from this species, which makes it an excellent marker compound for quality control. It is worth mentioning that the neo-clerodanes type compounds detected in the infusion were **1** and **8a,b**, but not the minor diterpenoids we previously isolated from the organic extract of the plant [8].

The analytical method was validated in terms of precision, accuracy, linearity and recovery according to the Q2 (R1) guideline published by the International Conference on Harmonisation (ICH) [30]. The linearity of the system was tested using a concentration range of **1** between 5 to 100 μg/mL and was found to be linear (R^2^ = 0.9994 (UV) and 0.9921 (Electrospray Ionization Mass Spectrometry, ESI-MS) in the concentration range used. The CV was less than 0.13% at each concentration level analyzed. The limit of identification (LOD) and quantification (LOQ) values were 1.22 and 3.70 μg/mL, respectively for UV detection; and 0.60 and 1.82 μg/mL, respectively for ESI-MS detection.

The linearity of the method was tested by recovery assay; the linear regression equation were found to be y = 5636.35x − 4337.55 (UV detection) and y = 5252.80x + 64,101.76 (ESI-MS detection). The recovery ranges for the standard were expressed as the concentration detected as a percentage of the expected concentration and were found in the ranges of 100.7–101.7 % for UV detection and 83.7–95.7% for ESI-MS detection. The reproducibility and repeatability of the analytical method were evaluated in terms of the intermediate precision by analyzing 6 replicates of the stock solution (50 μg/mL) in two different days. The relative SD (RSD; *n* = 6) was calculated for each sample evaluated. The results indicated that their chromatographic patterns were similar showing the presence of amarisolide (**1**). The CV values for accuracy were less than 0.11%. Subsequently, compound **1** was quantified, and the mean concentration calculated was 116 mg/g in dry matter.

## 3. Materials and Methods

### 3.1. General Procedures

IR spectra were recorded using a Bruker Tensor 27 FT-IR spectrophotometer (Bruker Corp., Billerica, MA, USA). Optical rotations were recorded at the sodium d-line wavelength using a Perkin Elmer model 343 polarimeter at 20 °C (Perkin Elmer, MA, USA). NMR spectra were registered on a Bruker AVANCE III HD with TCI CryoProbe 700 H-C spectrometer at 700 MHz (^1^H) or 175 MHz (^13^C), using TMS as an internal standard (Bruker Corp., Billerica, MA, USA). DARTHRMS were acquired with a JEOL AccuTOF-DART JMS-T100LC (JEOL Ltd., Tokyo, Japan) spectrometer in positive mode. For GC-MS analyses, an Agilent 6890N series gas chromatograph coupled to a LECO (Laboratory Equipment Corporation) time-of-flight mass spectrometer detector (MS-TOF; Agilent Technology, Santa Clara, CA, USA) was used. UHPLC-MS analyses were performed on a Waters Acquity UHPLC-H® Class system (Waters, Darmstadt, Germany) equipped with a quaternary pump, sample manager, column oven and photodiode array detector (PDA) interfaced with an SQD2 single mass spectrometer detector with an electrospray ion source. Column chromatography (CC) was carried out on Sephadex LH-20 (GE Healthcare, IL, USA). Thin layer chromatographic (TLC) analyses were performed on silica gel 60 F_254_ plates (Merck, Darmstadt, Germany), or *C*_18_-silica gel matrix plates Analtech plates (Merck, Darmstadt, Germany), visualization of the plates was carried out using an (NH_4_)_4_Ce(SO_4_)_4_ (10%) solution in H_2_SO_4_. Reagent-grade EtOAc, CHCl_3_, CH_2_Cl_2_ and MeOH were purchased from J.T. Baker (Avantor Performance Materials, PA, USA). MeCN, MeOH and water LC-MS or HPLC grades were purchased from Honeywell Burdick & Jackson (Morristown, NJ, USA). All other analytical grade solvents and reagents were obtained from various commercial sources. Amarisolide (**1**) was isolated from the species *S. amarissima* in the present study. The purity was determined to be more than 98% by UHPLC-MS.

### 3.2. Plant Material

*Salvia amarissima* was collected in Capulálpam de Méndez, Ixtlán de Juárez, Oaxaca, in January 2019 (Sa-Batch 1 (fresh) and Sa-Batch 2 (air-dried)). A voucher specimen (Number 1502277) was deposited at the National Herbarium of Mexico (MEXU), Instituto de Biología, UNAM. R. Torres-Colin achieved the botanical identification of the plant. The plant was air-dried and ground to a fine powder (2 mm) in a Thomas Wiley Model 4 Mill.

### 3.3. Extracts and Essential Oils Preparation

AE from *S. amarissima* (dried aerial parts) was prepared with 250 mL of boiling water and 12.5 g of the crude drug for 30 min. After filtration, the aqueous extract was concentrated in vacuo to obtain 0.1 g of a green residue. This process was repeated as necessary to prepare 10 g of AE. The ethyl acetate soluble fraction was prepared via partitioning with EtOAc (3 × 250 mL) from the aqueous extract. The resulting organic phase was dried over anhydrous sodium sulfate and concentrated in vacuo to yield 130 mg of a brown residue (yield 1.0%).

EOs were prepared from fresh (Sa-batch 1) and air-dried (Sa-batch 2) plant material (200 g in 1.5 L of distilled water) by hydrodistillation in a modified Clevenger type apparatus for 3 h. In both cases, the hydrodistilled was extracted with CH_2_Cl_2_ (3 × 2 L). The resulting organic phases were dried over Na_2_SO_4_ and concentrated in vacuo to yield an oily yellow residue (0.040 g, yield 0.02% in the case of the fresh material, and 0.063 g, yield 0.03% for the dried plant). All samples were stored at −4 °C until chemical analysis.

### 3.4. Separation of Active Compounds from the Ethyl Acetate Soluble Fraction

The ethyl acetate soluble fraction (100 mg) was subjected to CC on Sephadex LH-20 (400 g) using MeOH as eluent; fractions were pooled into 20 secondary fractions (F_1_–F_20_) according to their TLC profiles. From fraction F_16_ (73 mg) crystallized 60 mg of amarisolide (**1**). Preparative RP-TLC of fraction F_20_ (10 mg) yielded 1.2 mg of 5,6,4′-trihydroxy-7,3′-dimethoxyflavone (**2**) and 4.2 mg of 6-hydroxyluteolin (**3**).

### 3.5. Separation of Active Compounds from AE

AE (4.3 g) was fractionated via CC on Sephadex LH-20 using a gradient system of methanol–water (water 40–100%); this process gave 12 secondary fractions (AE_1_–AE_12_). From fraction AE_9_ (35 mg) crystallized 30 mg of rutin (**4**; m.p. 241–242 °C). From fraction AE_11_ (15 mg) crystallized 5 mg of isoquercitrin (**6**). Fraction AE_6_ (220 mg) was further purified on a Sephadex CC, eluting with MeOH, to yield 16 mg of rosmarinic acid (**5**). Preparative TLC on silica gel [ethyl acetate-methanol (85:15)] of fraction AE_5_ (11 mg) afforded 1 mg of pedalitin (**7**). Finally, preparative RP-TLC of AE_6_ (10 mg, MeOH) afforded 4 mg of an epimeric mixture of amarisolide G (**8a**,**b**).

Amarisolide G (**8a,b**): White solid; m.p. 133–135 °C. ${\left[\alpha \right]}_{D}^{20}$ = –149 (c = 1 mg/mL, MOH). UV (MeOH): λ_max_ (log ε) 206 (0.612) nm. IR (KBr): ν_max_ 3365, 1749 cm^−1^. ECD (c 0.2 mM, MeOH): λ_max_ (Δε) 212 (–5.21), 250 (–4.65) nm. ^1^H and ^13^C-NMR: see Table 1. HRESIMS: *m/z* 525.2317 [M + H]^+^ (calc. 525.2330 for C_26_H_37_O_11_).

### 3.6. Enzymatic Hydrolysis of ***8a,b***

Compound **8a**,**b** (2 mg) was mixed with β-glucosidase (2 mg, Sigma-Aldrich, MO, USA) in phosphate buffer solution (2 mL, 100 mM at pH 7); and kept at 40 °C for 15 days. Subsequently, the reaction mixture was extracted with CHCl_3,_ and the aqueous phase was concentrated to dryness and subjected to TLC analysis. d-Glucose was identified by comparison of the retention factor and optical rotation value with those of the authentic sample.

### 3.7. UHPLC-MS Analysis and Method Validation

The analytical method (Figure 5) was developed using an Acquity UHPLC^®^ BEH Shield C_18_ column (2.1 × 100 mm, 1.7 μm) at 40 °C. The mobile phase consisted of (A) water (0.1% formic acid) and (B) acetonitrile with a linear gradient elution program: 0–10 min, 20–100% (B); 10–10.5 min, 20% (B); 10.5–13 min, 20% (B). The flow rate was set to 0.3 μL/min, and the sample injection volume was 3.0 μL; detection was achieved with a PDA detector at 270 nm. For the identification of compounds, each sample was analyzed with the electrospray ion source operating in both positive (ESI^+^) and negative (ESI^−^) ionization modes. The ESI-MS conditions consisted of capillary voltage at 3.0 or 2.5 kV in positive and negative ion modes, respectively; dry heater temperature 150 °C; and nitrogen as the sheath gas flow. MS spectra were obtained within a mass range of *m*/*z* 100–1000 using nitrogen as the collision gas. The MassLynk software (version 4.1) was used to control of the UHPLC-MS system and for data acquisition and processing.

The method was validated according to the ICH guidelines [30]. For linearity, amarisolide (**1**) was accurately weighed and dissolved in dioxane-methanol (*v*/*v*, 1:1) to prepare stock solution at a final concentration of 1 mg/mL. Six working solutions in the range of 5–100 μg/mL for the standard were prepared from serial dilutions from the stock solution. Each concentration was prepared in sextuplicate. The linearity was assessed estimating the slope, y-intercept and coefficient of determination (R^2^) using the least squares method. Limits of detection (LOD) and quantification (LOQ) for the standard were determined at signal-to-noise (S/N) ratios of 3 and 10, respectively. Recovery experiments were carried out to evaluate the accuracy, assaying independently three amounts equivalent to 50 (ca. 10 μg/mL), 100 (ca. 50 μg/mL) or 125% (ca. 75 μg/mL). At each level, compound **1** was added simultaneously to the ethyl acetate soluble fraction (50 μg/mL). Each sample was injected twice and analyzed according to the method previously described. The mean percentage recovery for the standard was found to be between 98 and 102% by means of Fisher’s F test [30]. Finally, the repeatability and inter-day precision was evaluated by testing six identical samples according to the above described method on two consecutive days and by two different analysts by triplicate. The relative standard deviation (RSD) was calculated for each determination as a measure of precision and repeatability.

### 3.8. GC-MS Analysis of the Essential Oils

For GC-MS analyses, compounds were separated on a DB-5 capillary column (Supelco, Bellefonte, PA, USA) with the following temperature program: oven temperature was programmed from 40 to 260 °C at 4 °C/min during 20 min, and finally up to 340 °C for 20 min isothermally; injector and MS transfer line temperatures were set at 200 and 300 °C, respectively; Helium was used as the carrier gas at a constant flow rate of 1 mL/min; split ratio, 1:20. A mixture of the homologous series of n-alkanes (C_8_–C_20_) in CH_2_Cl_2_ was directly injected into the GC under the above temperature program, in order to calculate the linear retention indices (R_I_). All mass spectra were acquired in EI mode (scan range *m/z* 40–400, ionization energy 70 eV). The components were identified using retention index (R_I_) of peaks in the chromatogram [31,32] and by comparison of their mass spectra with those of standard library data (NIST) of the GC-MS system and literature data or with those of authentic samples available commercially. All determinations were performed in triplicate.

### 3.9. Protein Tyrosine Phosphatase 1B Inhibition Assay

The expression and purification of hPTP-1B was performed as previously described [33]. Aqueous extract (AE), ethyl acetate soluble fraction, pure compounds and positive control were dissolved in DMSO or MeOH or Tris buffer solution (Tris-HCl, 20 mM, pH 7). Aliquots of 0–10 μL of testing materials (triplicated) were incubated for 5 min with 20 μL of enzyme stock solution in Tris-HCl (22 nM). After incubation, 10 μL of p-nitrophenylphosphate (pNPP; 5 mM) was added and further incubated for 15 min at 25 °C; then, the absorbance was determined (λ_max_ 415 nm). For all samples, the inhibitory activity was determined as a percentage in comparison to the blank (Tris-HCl) according to the following equation:(1)$$\%\;\mathrm{PTP}1B=\left(1-\frac{A_{415t}}{A_{415C}}\right)\times 100$$
where % PTP-1B is the percentage of inhibition, *A*_415t_ is the corrected absorbance of the extracts, fraction, or compounds under testing (*A*_415end_ − *A*_415initial_) and *A*_415C_ is the absorbance of the blank (*A*_415end blank_ − *A*_415initial blank_). The IC_50_ was calculated by regression analysis, using the following equation:(2)$$\%\;Inhibition=\frac{A_{100}}{1+{\left(\frac{I}{IC_{50}}\right)}^{s}}$$
where *A*_100_ is the maximum inhibition, *I* is the inhibitor concentration, *IC*_50_ is the concentration required to inhibit the activity of the enzyme by 50% and *S* is the cooperative degree.

### 3.10. Docking Studies

To perform the docking at the catalytic site, the PTP1B-INZ complex corresponding to the PDB 1G7F was used, which has a resolution of 1.8 A. For the allosteric site, the PDB 1T49 (resolution of 1.9) was used. The two PDBs used were selected considering the resolution and that they had co-crystallized ligands at the sites of interest. All compounds were built using the HyperChem 8.0 release program and optimized geometrically using the Gaussian 09 program, revision A.02 (Gaussian Inc., Wallingford, CT, USA) at DFT B3LYP/3-21G level of theory. The protein and ligands were further prepared using the utilities implemented by AutoDockTools 1.5.4 (http://mgltools.scripps.edu/). The protein was adding polar hydrogen atoms, Kollman united-atom partial charges, and to the ligands computing Gasteiger–Marsilli formalism charges, rotatable groups which were assigned automatically as were the active torsions. Blind docking was carried out using AutoDock Vina version 2.0 [34]. The root mean square deviation (RMSD) values were obtained by comparing the best pose generated in AutoDock. The initial parameters used for the active site were a grid box size was 42 Å × 40 Å × 40 Å in the x, y and z dimensions and grid center 9.73, 18.00, 971 to 1G7F.PDB. For the allosteric site docking were a grid box size was 42 Å × 40 Å × 40 Å in the x, y and z dimensions and grid center 9.73, 18.00, 971 used 1T49.pdb. For both sites, the exhaustiveness was 25, and the ten best poses were obtained. The analysis of the docking was made with PyMol (Maestro, Schrödinger, LLC, New York, NY) [35]. 

### 3.11. Molecular Dynamics Simulation

All the structural complexes were verified, cleaned and ordered with the pdb4amber scrip before starting the preparation in order to generate suitable topologies from the LEaP module of AMBER 19 [36,37]. Each structure and complex was subjected to the following protocol: hydrogens and other missing atoms were added using the LEaP module with the leaprc.protein.ff19SB parameter set; Cl^−^or K^+^ counterions were added to neutralize the system; the complexes were then solvated in an octahedral box of explicit TIP3P model water molecules localizing the box limits at 12 Å from the protein surface. Molecular dynamic simulations were performed at 1 atm and 315 K, maintained with the Berendsen barostat and thermostat, using periodic boundary conditions and particle mesh Ewald sums (grid spacing of 1 Å) for treating long-range electrostatic interactions with a 10 Å cutoff for computing direct interactions. The SHAKE algorithm was used to satisfy bond constraints, allowing the employment of a two fs time step for the integration of Newton’s equations as recommended in the Amber package [36,38]. Amber leaprc.protein.ff19SB force field [39] parameters were used for all residues. All calculations were made using Graphics Processing Units (GPU) accelerated MD engine in AMBER (pmemd.cuda), a program package that runs entirely on CUDA® (Compute Unified Device Architecture)-enabled GPUs [40]. The protocol consisted of performing a minimization of the initial structure, followed by 50ps heating and pressure equilibration at 315 K and 1.0 atm pressure, respectively. Finally, the system is equilibrated with 500ps before starting the production of MD. The production of the MD consisted of 100 ns for each complex. Frames were saved at ten ps intervals for subsequent analysis. All analyses were done using CPPTRAJ [41] part of AMBER19 utilities and OriginPro 9.1. The calculations of RMSD and Root Mean Square Fluctuations (RMSF) were made, considering the C, CA and N. The charts were built with OriginPro 2018 SR1, and the trends were adjusted with the function processing smooth (method lowess span). VMD and PyMOL [35] were used to visualize and create the images from the MD.

### 3.12. Chemoinformatic Properties of Compounds ***1, 7*** and ***AU***

The biological and chemoinformatic properties of compounds **1**, **7** and **AU** were explored using the servers Swiss TargetPrediction (http://www.swisstargetprediction.ch/index.php), Molinspiration (http://www.molinspiration.com/cgi-bin/properties) and Osiris Property Explorer server (http://che minformatics.ch/propertyExplorer/) [29,42].

## 4. Conclusions

The AE of *S. amarissima* contains rutin (**4**) and rosmarinic acid (**5**), which inhibit intestinal glucose absorption, promote glucose uptake in muscle cells and suppress insulin-resistance, among other effects. On the other hand, pedalitin (7), which behave in silico as an allosteric inhibitor of PTP-1B, could contribute to its overall antidiabetic action via α-glucosidase and selective PTP-1B inhibition, and other mechanisms yet to be determined. The overall action of AE could be attained via network pharmacology, synergism and or polypharmacology. Altogether, our studies on *S. amarissima* tend to support its medicinal use for the treatment of diabetes in Mexican folk medicine. The chromatographic analyses developed and validated in this study will allow the development of a pharmacopeic monograph, and generate standardized preparations of this very Mexican plant. The analytical UHPLC method was suitable for its intended purpose, the quantification of amarisolide (**1**) according to the Q2 (R1) guideline. Overall, the scientific information generated for this plant will contribute to its rational use in Mexican folk medicine. Like many other New World melittophilous *Salvia* species, *S. amarissima* is a rich source of bioactive compounds.

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
