# Peer review of "Flavonoids and Terpenoids with PTP-1B Inhibitory Properties from the Infusion of Salvia amarissima Ortega"

_molecules, 2020, doi:10.3390/molecules25153530_

Round 1

Reviewer 1 Report

PTP-1B  in title should be spelled out in full as protein tyrosine phosphatase 1B. The significance of  PTP-1B inhibitory activity should be mentioned briefly in the abstract.

The chemistry part of the manuscript is good.

The significance of the finding of PTP-1B inhibitory activity in compounds from the Infusion of Salvia amarissima 4 Ortega should be dealt with in greater detail in the Results and Discussion. As it stands there is minimal discussion.

Lines 226 and 232 species names have to be in italics.

Minor errrors in English need to be corrected e.g. line 226: in January, NOT "on January"

Author Response

Reviewer 1:

a.-Q1: The significance of the finding of PTP-1B inhibitory activity in compounds from the Infusion of Salvia amarissima  Ortega should be dealt with in greater detail in the

We addressed in detail this issue indicating what could be the contibutions of the finding in the overall actions, considering polypharmacology, network pharmacology, and even synergism. 

b.-Q2:  Lines 226 and 232 species names have to be in italics.

This was corrected.

c.-Q3: Minor errrors in English need to be corrected e.g. line 226: in January, NOT "on January".

They were corrected.

Reviewer 2 Report

The article entitled "Flavonoids and Terpenoids with PTP-1B Inhibitory Properties from the Infusion of Salvia amarissima Ortega" has little scientific innovation, despite being well written and reasoned the article needs additional data so that it can be accepted.

Minor comments

  1. I suggest that the authors increase the introduction and make a schematic flowchart to facilitate the visualization of the methodological steps;
  2. I suggest that figures 2 and 3 be inserted in the supplementary material;
  3. What is the justification for authors to use the Autodock Tools package v1.5.4/AutoDock v1.1.2.15 instead of the AutoDock Vina software (The Scripps Research Institute, La Jolla, CA- Trott, O.; Olson, A.J. AutoDock Vina: improving the speed and accuracy of docking with a new scoring function, efficient optimization, and multithreading. J Comput Chem 2009, 31, 455-461, doi:10.1002/jcc.21334)?
  4. I suggest that the authors include in silico predictions using the Swiss Target Prediction website (https://www.swisstargetprediction.ch) comparing these results with the molinspirtation server.

Major revisions:

  1. Authors should increase their discussions, creating a topic about structure and activity relationship comparing with some commercial drug that should be included in the study;
  2. What was the selection criterion for choosing the PDB ID used in the study of molecular docking, as such information isn't clear in the form in which the manuscript was presented and needs modifications and rewriting?.
  3. The authors do not mention in the docking study the RMSD value. What was the criterion used to select the region of activity?. Moreover, authors should reproduce a Figure of superpositions of crystallographic ligands poses with the calculated poses or use computational tools to search for the likely region of the active site (add - supplementary material). One of them must be in red and one in green. Also, these are the most common colours for colour blindness, for example.
  4. While docking was shown to reproduce crystallographic binding modes and should be shown that docking to at least one of the protein structures can reproduce experimental binding energies, or at least that docking energies can accurately rank ligands according to their observed activities; at no stage in relation to the data are shown. This is crucial, as the authors are comparing binding energies between different ligands. From this perspective, the work presented is spurious and needs to be revised, if not entirely replaced/refocused.
  5. I suggest that the authors perform a selection of the receptor-ligand complex (target protein) taking into account (1) the structural similarity (visual inspection) of the ligands to the template scaffold structure, regarding the presence of the pharmacophore regions; followed by (2) the overlap of the ligands with the template structure, in relation to the overlap similarity values, see more details: (a) Costa, J.S.; Costa, K.S.L.; Cruz, J.V.; Ramos, R.S.; Silva, L.B.; Brasil, D.S.B.; Tomich, C.S.; Rodrigues, C.B.; Macêdo, W.J.C. Virtual screening and statistical analysis in the design of new caffeine analogues molecules with potential epithelial anticancer activity. Curr. Pharm. Des. 2018, 24, 576–594. (b) Borges, R.S.; Palheta, I.C.; Ota, S.S.B.; Morais, R.B.; Barros, V.A.; Ramos, R.S.; Silva, R.C.; Costa, J.S.; Silva, C.H.T.P.; Campos, J.M.; et al. Toward of Safer Phenylbutazone Derivatives by Exploration of Toxicity Mechanism. Molecules 2019, 24, 143.
  6. The authors do not perform Delta G calculations to investigate whether the reaction is spontaneous or not spontaneous. Such information should be related to the in silico data obtained. The section must be fully revised to improve understanding of the technical details and the comparisons between experimental and predicted binding affinities. Moreover, the docking had the ability to reproduce the experimental binding affinities. Values can be calculated from experimentally determined inhibition constants (Ki) found in the PDBs or determined experimentally as in this study, according to Equation: ΔG = R.T.lnKi [Cera, E.D. Thermodynamic Theory of Site-Specific Binding Processes in Biological Macromolecules; Cambridge University Press: Cambridge, 1995; Gohlke, H.; Klebe, G. Approaches to the description and prediction of the binding affinity of small-molecule ligands to macromolecular receptors. Angewandte Chemie (International ed. in English) 2002, 41, 2644-2676].
  7. Molecular docking studies - The use of a single pose is not enough (How many were the run and what was the criterion of molecular docking for runs?). It is known the "low resolution" of standard docking score functions. A higher level of theory must be implemented to compute interactions energies. The authors need to detail the in silico protocols, such as: the X, Y and Z coordinates of the GRID BOX or active site was not clear (insert in Table form – see paper “Identification of Novel Chemical Entities for Adenosine Receptor Type 2A Using Molecular Modeling Approaches. Molecules 2020, 25(5), 1245. https://doi.org/10.3390/molecules25051245 “.
  8. Molecular docking study informing distances and/or bond length is not enough to elucidate the mechanism of action. I suggest that additional molecular dynamics studies be conducted to investigate the molecular stability of the compounds always using a commercial compound, in order to compare such in silico results. See more details: (a) Potential inhibitors of the enzyme acetylcholinesterase and juvenile hormone with insecticidal activity: study of the binding mode via docking and molecular dynamics simulations. Journal of Biomolecular Structure & Dynamics, v. 37, p. 1-23, 2019. https://doi.org/10.1080/07391102.2019.16881924.  (b) Identification of Potential Inhibitors from Pyriproxyfen with Insecticidal Activity by Virtual Screening. PHARMACEUTICALS, v. 12, p. 20, 2019. Pharmaceuticals 2019, 12(1), 20; https://doi.org/10.3390/ph12010020  (c) Hierarchical Virtual Screening of Potential Insectides Inhibitors of Acetylcholinesterase and Juvenile Hormone from Temephos. PHARMACEUTICALS, v. 12, p. 61, 2019. Pharmaceuticals 2019, 12(2), 61; https://doi.org/10.3390/ph12020061 (d) Studies of NMR, molecular docking, and molecular dynamics simulation of new promising inhibitors of cruzaine from the parasite Trypanosoma cruzi. Med. Chem. Res. 2019, 28, 246–259 (e) Pinto, V.; Araújo, J.; Silva, R.; da Costa, G.; Cruz, J.; De, A.; Neto, M.; Campos, J.; Santos, C.; Leite, F.; et al. In Silico Study to Identify New Antituberculosis Molecules from Natural Sources by Hierarchical Virtual Screening and Molecular Dynamics Simulations. Pharmaceuticals 2019, 12, 36. (f) Molecular dynamics simulation and binding free energy studies of novel leads belonging to the benzofuran class inhibitors of Mycobacterium tuberculosis Polyketide Synthase 13. J. Biomol. Struct. Dyn. 2019, 37, 1616–1627. (g) Oil from the fruits of PterodonemarginatusVog.: A traditional anti-inflammatory. Study combining in vivo and in silico. J. Ethnopharmacol. 2018, 222, 107–120.  (h) Computational design of new protein kinase 2 inhibitors for the treatment of inflammatory diseases using QSAR, pharmacophore-structure-based virtual screening, and molecular dynamics. J. Mol. Model. 2018, 24, 225-241.  (i) In Silico Evaluation of Ibuprofen and Two Benzoylpropionic Acid Derivatives with Potential Anti-Inflammatory Activity. Molecules 2019, 24, 1476;
  9. The experimental design is straightforward; the data are strong and support the conclusions. However, such questions raised here need to be better clarified in order to be accepted the manuscript. I firmly believe that the findings reported here will have a major impact on the field of molecular modeling, especially in the search for new inhibitors.

Author Response

Reviewer 2:

Before answering the queries of reviewer 2, I have to make clear that it was not an intended theoretical chemistry paper. Please see the aims of the work. The ides was only to predict potential attachment sites of each type of compound.

a.-Q1: I suggest that the authors increase the introduction and make a schematic flowchart to facilitate the visualization of the methodological steps.

Well, I think the manuscript is very concise. I appreciate the suggestion, but we preferred our style of presenting this type of paper.

b.-Q2: I suggest that figures 2 and 3 be inserted in the supplementary material.

Done

c.-Q3: What is the justification for authors to use the Autodock Tools package v1.5.4/AutoDock v1.1.2.15 instead of the AutoDock Vina software (The Scripps Research Institute, La Jolla, CA- Trott, O.; Olson, A.J. AutoDock Vina: improving the speed and accuracy of docking with a new scoring function, efficient optimization, and multithreading. J Comput Chem 2009, 31, 455-461, doi:10.1002/jcc.21334)?

The results obtained with both programs did not show significant differences, so in this new version, however, we used Autodock vina. We have done the docking again directed to the active and allosteric site, and based on this, we chose the pdb that was co-crystallized ligands in these sites, and that also had a resolution less than two angstroms.

d.-Q4: I suggest that the authors include in silico predictions using the Swiss Target Prediction website (https://www.swisstargetprediction.ch) comparing these results with the molinspirtation server.

Probably for the next paper.

e.-Q5 Authors should increase their discussions, creating a topic about structure and activity relationship comparing with some commercial drug that should be included in the study.

Probably, the referee does not that there is not any drug targeting PTP-1B in the market. There is not enough data to propose any activity-structure relationship.

f.-Q6: What was the selection criterion for choosing the PDB ID used in the study of molecular docking, as such information isn't clear in the form in which the manuscript was presented and needs modifications and rewriting?.

We have done the docking again directed to the active and allosteric sites, and based on this, we chose the pdb targets was co-crystallized ligands in these sites, and that also had a resolution less than two angstroms.

g.-Q6: The authors do not mention in the docking study the RMSD value. What was the criterion used to select the region of activity? Moreover, authors should reproduce a Figure of superpositions of crystallographic ligands poses with the calculated poses or use computational tools to search for the likely region of the active site (add - supplementary material). One of them must be in red and one in green. Also, these are the most common colours for colour blindness, for example.

We docked at two sites (catalytic and allosteric sites), and affinity (binding energy) was the criterion for establishing the most likely binding site. In Figure 4, we superimposed the resulting docking poses with the structure of the co-crystallized ligand. In the same way, we established standard colors to the compounds for a better appreciation.

h.-Q7: While docking was shown to reproduce crystallographic binding modes and should be shown that docking to at least one of the protein structures can reproduce experimental binding energies, or at least that docking energies can accurately rank ligands according to their observed activities; at no stage in relation to the data are shown. This is crucial, as the authors are comparing binding energies between different ligands. From this perspective, the work presented is spurious and needs to be revised, if not entirely replaced/refocused.

We did it with two compounds, we wanted to analyzed. As I said, the idea was to predict the site of binding.

i.-Q8: I suggest that the authors perform a selection of the receptor-ligand complex (target protein) taking into account (1) the structural similarity (visual inspection) of the ligands to the template scaffold structure, regarding the presence of the pharmacophore regions; followed by (2) the overlap of the ligands with the template structure, in relation to the overlap similarity values, see more details.

Two PTP1B-Ligand complexes were selected, taking into account the binding sites and the structure of the co-crystallized ligands 892 and INZ.

j.-Q9: The authors do not perform Delta G calculations to investigate whether the reaction is spontaneous or not spontaneous. Such information should be related to the in silico data obtained. The section must be fully revised to improve understanding of the technical details and the comparisons between experimental and predicted binding affinities. Moreover, the docking had the ability to reproduce the experimental binding affinities. Values can be calculated from experimentally determined inhibition constants (Ki) found in the PDBs or determined experimentally as in this study, according to Equation: ΔG = R.T.lnKi.

Thanks for the suggestion, we now include estimated bond energy and additionally calculate a Ki from the docking using the equation ΔG = RTlnKi (Table 4)

k.-Q10: Molecular docking studies - The use of a single pose is not enough (How many were the run and what was the criterion of molecular docking for runs?). It is known the "low resolution" of standard docking score functions. A higher level of theory must be implemented to compute interactions energies. The authors need to detail the in silico protocols, such as: the X, Y and Z coordinates of the GRID BOX or active site was not clear.

We have rewritten the docking section on materials and methods, where the procedure is detailed.

lQ-11: Molecular docking study informing distances and/or bond length is not enough to elucidate the mechanism of action. I suggest that additional molecular dynamics studies be conducted to investigate the molecular stability of the compounds always using a commercial compound, in order to compare such in silico results.

With the docking analysis, we do not pretend to determine any mechanism. We d’ont feel at the present stage that it is necessary to carried out Molecular Dynamic Simulations. We are more involved to find out in vivo if the signaling cascade of insulin is really shut down by compound 7. After that, the study this referee is proposing will be interesting. Please it is not the purpose of this work.

Reviewer 3 Report

The authors present a well-structured work with a significant amount of results. My only concern is in the conclusions section. Neo-clerodane diterpenes are unlikely to participate in the antidiabetic activity found in the traditional use of the plant. They have a lower inhibitory activity than the other tested compounds and moreover they will surely be in very low concentrations in the traditional infusion.

Although they have shown a possible inhibitory effect, this cannot be associated with the traditional use of the infusion.

In addition, the lack of the quantification data of the compounds within the extract does not allow to identify the major contributor to the inhibitory activity.

In the conclusion the names of the plants are not in italics.

Author Response

a.-Q1: Although they have shown a possible inhibitory effect, this cannot be associated with the traditional use of the infusion.

Yes, it can be associated, because in a previous work we tested the infusion in vivo.  Any way we have added the work “might be correlated”.

b.-Q2: In addition, the lack of the quantification data of the compounds within the extract does not allow to identify the major contributor to the inhibitory activity.

In this regard, we presented a chromatogram that gives anyone an idea of the concentration of each compound in the traditional preparation. Compound 1 was quantified. Beyond that, I am under the impression that the referee is not familiar with synergism, polypharmacology or network pharmacology. Thus, we add the following statement:

“The results of the PTP-1B are significant. They suggest that the traditional preparation of S. amarissima, with hypoglycemic and antihyperglycemic properties demonstrated in vivo [7] (i.e., the overall action), contains compounds such as 1-8a,b that might weakly target different proteins (i.e., PTP-1B and others) within the same signaling network thus shutting insulin signaling cascade process by network pharmacology. It is also possible that compounds 1-6 and 8a,b, with weaker activity than compound 7, altogether put forth a biochemical effect by synergism (i.e., a synergy between weakly active compounds against PTP-1B). Finally, molecules like compounds 1, 4, 5, and 7 can exert their action binding different targets such as PTP-1B and a-glycosidases, among others (polypharmacology). The facts that rutin (5) [23] and rosmarinic acid (7) [21] are multitarget antidiabetic compounds, and compounds 1 and 7 inhibited a-glycosidases in vivo [7] strengthen any of these possibilities.”

c.-Q3: In the conclusion the names of the plants are not in italics.

Now the name of the plant is in italic.

Round 2

Reviewer 2 Report

The article entitled "Flavonoids and Terpenoids with PTP-1B Inhibitory Properties from the Infusion of Salvia amarissima Ortega#" has little scientific innovation, despite being well written and reasoned the article needs additional data so that it can be accepted.

  1. In line 183, check rosmarinic acid (7), because the numbering (7) appears is correct?

General comments:

  1. Regarding the comment that the author states “I have to make clear that it was not an intended theoretical chemistry paper. Please see the aims of the work. The ides was only to predict potential attachment sites of each type of compound”. I believe that from the moment that an in silico study is inserted to justify a certain mechanism of action, the work has a theoretical-experimental objective. Therefore, I suggest the authors read the recently published article (2020) and include it in the introduction if possible “Assessment of the hypoglycemic effect of Bixin in alloxan-induced diabetic rats: in vivo and in silico studies”, Journal of Biomolecular Structure and Dynamics, https://doi.org/10.1080/07391102.2020.1724567

  1. In the previous comment the authors did not follow “I suggest that the authors increase the introduction and make a schematic flowchart to facilitate the visualization of the methodological steps”. Therefore, I again request the construction of the flowchart to facilitate reading similar to the recently published article, see - Identification of Novel Chemical Entities for Adenosine Receptor Type 2A Using Molecular Modeling Approaches. Molecules 2020, 25(5), 1245.

  1. Regarding the authors' response to the questioning “I suggest that the authors include in silico predictions using the Swiss Target Prediction website (https://www.swisstargetprediction.ch) comparing these results with the molinspirtation server” – I believe that the article would be more robust, since it is based on a very simple methodology to be carried out in a few minutes.

  1. Another question asked was “Authors should increase their discussions, creating a topic about structure and activity relationship comparing with some commercial drug that should be included in the study”. I think the authors did not understand very well the structure-activity relationship that was requested - if they do not have a commercial compound, the authors can perform the Structure-Activity Relationship of the promising molecule (SAR) with the control compound used in the experiment (ursolic acid) among selected compound more activity (compound 7, pedalitin). Therefore, I request that the authors consult the articles below to facilitate understanding:
  • Ferreira, E.F.B.; Silva, L.B.; Costa, G.V.; Costa, J.S.; Fujishima, M.A.T.; Leão, R.P.; Ferreira, A.L.S.; Federico, L.B.; Silva, C.H.T.P.; Rosa, J.M.C.; Macêdo, W.J.C.; Santos, C.B.R. Identification of New Inhibitors with Potential Antitumor Activity from Polypeptide Structures via Hierarchical Virtual Screening. Molecules 2019, 24, 2943;
  • Potential inhibitors of the enzyme acetylcholinesterase and juvenile hormone with insecticidal activity: study of the binding mode via docking and molecular dynamics simulations. Journal of Biomolecular Structure & Dynamics, v. 37, p. 1-23, 2019. https://doi.org/10.1080/07391102.2019.16881924;
  • Identification of Potential Inhibitors from Pyriproxyfen with Insecticidal Activity by Virtual Screening. PHARMACEUTICALS, v. 12, p. 20, 2019. Pharmaceuticals 2019, 12(1), 20; https://doi.org/10.3390/ph12010020
  • Hierarchical Virtual Screening of Potential Insectides Inhibitors of Acetylcholinesterase and Juvenile Hormone from Temephos. PHARMACEUTICALS, v. 12, p. 61, 2019. Pharmaceuticals 2019, 12(2), 61; https://doi.org/10.3390/ph12020061
  • Cruz, J.V.; Serafim, R.B.; da Silva, G.M.; Giuliatti, S.; Rosa, J.M.C.; Neto, M.F.A.; Leite, F.H.A.; Taft, C.A.; da Silva, C.H.T.P.; Santos, C.B.R. Computational design of new protein kinase 2 inhibitors for the treatment of inflammatory diseases using QSAR, pharmacophore-structure-based virtual screening, and molecular dynamics. J. Mol. Model. 2018, 24, 225.

  1. The authors do not mention in the docking study the RMSD value in this new version, such information must follow the molecular docking protocols, see further information for clarification by the authors in the article Ramírez, D.; Caballero, J. Is It Reliable to Take the Molecular Docking Top Scoring Position as the Best Solution without Considering Available Structural Data? Molecules 2018, 23, 1038, if possible the authors should cite that article in their discussion. Selected docking solutions are good, acceptable and bad according to above mentioned RMSD criteria. It is clear that an RMSD < 2.0 Å corresponds to good docking solutions. On the other hand, docking solutions with RMSD between 2.0 and 3.0 Å deviate from the position of the reference, but they keep the desired orientation. Finally, docking solutions with RMSD > 3.0 Å are completely wrong.

  1. Regarding the first comment in the review round “Molecular docking study informing distances and/or bond length is not enough to elucidate the mechanism of action. I suggest that additional molecular dynamics studies be conducted to investigate the molecular stability of the compounds always using a commercial compound, in order to compare such in silico results” – The authors' responses worry me, because during the pharmacological process there are several additional questions, such as the stability of the compound within the active site, as well as the pharmacokinetic and toxicological processes. Hence the importance of in silico predictions, as everything is mathematical models of predictions whether in vivo, in vitro or in silico.

  1. In the summary, the following is reported: “revealed that germacrene D (15) and b-selinene (16), followed by b-25 caryophyllene (13) and spathulenol (17) were their major components”. However, the authors claim “We are more involved to find out in vivo if the signaling cascade of insulin is really shut down by compound 7”. Segundo o artigo “among the flavonoids tested, the most active component was pedalitin (7) with an IC50 of 62.0 ± 4.1µM”, why the compounds germacrene D (15) and b-selinene (16), followed by b-25 caryophyllene (13) and spathulenol (17) have not been tested? Such information should be further clarified in the text.

  1. The experimental design is straightforward; the data are strong and support the conclusions. However, such questions raised here need to be better clarified in order to be accepted. I firmly believe that the findings reported here will have a major impact in the scientific field.

Author Response

We revised the manuscript according to the suggestions of the reviewer:
Q1: Regarding the comment that the author states “I have to make clear that it was not an intended theoretical chemistry paper. Please see the aims of the work. The ides was only to predict potential attachment sites of each type of compound”. I believe that from the moment that an in silico study is inserted to justify a certain mechanism of action, the work has a theoretical-experimental objective. Therefore, I suggest the authors read the recently published article (2020) and include it in the introduction if possible “Assessment of the hypoglycemic effect of Bixin in alloxan-induced diabetic rats: in vivo and in silico studies”, Journal of Biomolecular Structure and Dynamics, https://doi.org/10.1080/07391102.2020.1724567
A1: Done.

Q2: In the previous comment the authors did not follow “I suggest that the authors increase the introduction and make a schematic flowchart to facilitate the visualization of the methodological steps”. Therefore, I again request the construction of the flowchart to facilitate reading similar to the recently published article, see - Identification of Novel Chemical Entities for Adenosine Receptor Type 2A Using Molecular Modeling Approaches. Molecules 2020, 25(5), 1245.
A2: Done

Q3: Regarding the authors' response to the questioning “I suggest that the authors include in silico predictions using the Swiss Target Prediction website (https://www.swisstargetprediction.ch) comparing these results with the
molinspirtation server” – I believe that the article would be more robust, since it is based on a very simple methodology to be carried out in a few minutes.
A3: Done.

Q4: Another question asked was “Authors should increase their discussions, creating a topic about structure and activity relationship comparing with some commercial drug that should be included in the study”. I think the authors did not understand very well the structure-activity relationship that was requested - if they do not have a commercial compound, the authors can perform the Structure-Activity Relationship of the promising molecule (SAR) with the control compound used in the experiment (ursolic acid) among selected compound more activity (compound 7, pedalitin). Therefore, I request that the authors consult the articles below to facilitate understanding.
A4: Done

Q5: The authors do not mention in the docking study the RMSD value in this new version, such information must follow the molecular docking protocols, see further information for clarification by the authors in the article Ramírez, D.; Caballero, J. Is It Reliable to Take the Molecular Docking Top Scoring Position as the Best Solution without Considering Available Structural Data? Molecules 2018, 23, 1038, if possible the authors should cite that article in their discussion. Selected docking solutions are good, acceptable and bad according to above mentioned RMSD criteria. It is clear that an RMSD < 2.0 Å corresponds to good docking solutions. On the other hand, docking solutions with RMSD between 2.0 and 3.0 Å deviate from the position of the reference, but they keep the desired orientation. Finally, docking solutions with RMSD > 3.0 Å are completely wrong.
A5: Done

Q6: Regarding the first comment in the review round “Molecular docking study informing distances and/or bond length is not enough to elucidate the mechanism of action. I suggest that additional molecular dynamics studies be conducted to investigate the molecular stability of the compounds always using a commercial compound, in order to compare such in silico results” – The authors' responses worry me, because during the pharmacological process there are several additional questions, such as the stability of the compound within the active site, as well as the pharmacokinetic and toxicological processes. Hence the importance of in silico predictions, as everything is mathematical models of predictions whether in vivo, in vitro or in silico.
A6: For this new version, molecular dynamics studies of the PTP-1B-ligand complexes were performed to evaluate molecular stability

Q-7: In the summary, the following is reported: “revealed that germacrene D (15) and selinene (16), followed by caryophyllene (13) and spathulenol (17) were their major components”. However, the authors claim “We are more involved to find out in vivo if the signaling cascade of insulin is really shut down by compound 7”. Segundo o artigo “among the flavonoids tested, the most active component was pedalitin (7) with an IC50 of 62.0 ± 4.1μM”, why the compounds germacrene D (15) and beta-selinene (16), followed by caryophyllene (13) and spathulenol (17) have not been tested? Such information should be further clarified in the text.
A7: I did not understand this question, what is segundo or artigo? Anyway, the components of the oil were not isolated. They were identified by a classical and well-known methodology using GC-MS. We were citing another article where these compounds were tested against PTP-1B. Nevertheless, we removed from this article the activity of the essential oil and just leave its composition, not previously reported as an identity test.
I hope that after these changes you find the manuscript suitable for publication.
With my personal best regards